# Uranium carbonate complexes demonstrate drastic decrease in stability at elevated temperatures

Alexander Kalintsev[1,2], Artas Migdisov [1✉], Christopher Alcorn[1], Jason Baker[1], Joël Brugger [2], Robert A. Mayanovic[3], Nadib Akram[3], Xiaofeng Guo[4], Hongwu Xu[1], Hakim Boukhalfa[1], Florie A. Caporuscio[1], Hari Viswanathan[1], Carlos Jove-Colon[5], Yifeng Wang[5], Edward Matteo[5] & Robert Roback[1]

Quantitative understanding of uranium transport by high temperature fluids is crucial for confident assessment of its migration in a number of natural and artificially induced contexts, such as hydrothermal uranium ore deposits and nuclear waste stored in geological repositories. An additional recent and atypical context would be the seawater inundated fuel of the Fukushima Daiichi Nuclear Power Plant. Given its wide applicability, understanding uranium transport will be useful regardless of whether nuclear power finds increased or decreased adoption in the future. The amount of uranium that can be carried by geofluids is enhanced by the formation of complexes with inorganic ligands. Carbonate has long been touted as a critical transporting ligand for uranium in both ore deposit and waste repository contexts. However, this paradigm has only been supported by experiments conducted at ambient conditions. We have experimentally evaluated the ability of carbonate-bearing fluids to dissolve (and therefore transport) uranium at high temperature, and discovered that in fact, at temperatures above 100 °C, carbonate becomes almost completely irrelevant as a transporting ligand. This demands a re-evaluation of a number of hydrothermal uranium transport models, as carbonate can no longer be considered key to the formation of uranium ore deposits or as an enabler of uranium transport from nuclear waste repositories at elevated temperatures.

[1] Earth & Environmental Division, Los Alamos National Laboratory, Los Alamos, NM, USA. [2] School of Earth, Atmosphere and Environment, Monash University, Clayton, VIC, Australia. [3] Department of Physics, Astronomy and Materials Science, Missouri State University, Springfield, MO, USA. [4] Department of Chemistry and Alexandra Navrotsky Institute for Experimental Thermodynamics, Washington State University, Washington, USA. [5] Sandia National Laboratory, Albuquerque, NM, USA. ✉email: artas65@gmail.com

Considering its low carbon footprint, high energy density and the emergence of novel technologies such as small modular reactors (SMRs), nuclear power is an attractive candidate for supplementing other low-carbon energy sources as the world continues to move away from fossil-fuel-based energy sources. Constructing and maintaining more reactors naturally brings with it the problem of producing enough fuel to keep them operational, and thus necessitates increased mining of and exploration for uranium resources (thorium technology notwithstanding). The majority of uranium ore deposits are typified by significant hydrothermal fluid activity occurring at temperatures ranging from 150 to 400 °C[1–3]. In most uranium ore deposits hydrothermal fluids leach uranium from moderately uranium-enriched source rocks (typically granites or gneisses though many sources have been postulated), and concentrate it along geochemical boundaries that induce its precipitation from the fluid[1,4–6]. It is worth-noting that the same processes that enable its transport in ore deposits also present a hazard in waste repositories. While most repository designs aren't expected to experience local rock temperatures above 100 °C[7–9], those intended for dual-purpose canisters could potentially reach local rock and fluid temperatures in excess of 200 °C[10–12]. Similar transportation processes also need to be considered in nuclear accidents such as the Fukushima Daiichi incident, where hot fuel assemblies were submerged by seawater permitting uranium migration into the local environment[13,14].

In general, uranium is most effectively transported by hydrothermal fluids when present in its hexavalent (U(VI)) oxidation state, which, as the uranyl ion ($UO_2^{2+}$), is highly soluble and forms stable complexes with a range of inorganic ligands, most commonly $Cl^-$, $SO_4^{2-}$, $OH^-$, and $CO_3^{2-}$[15,16]. Thus, the presence of such ligands serves as a principal way to enhance the uranium carrying capacity of hydrothermal fluids. While naturally dependent upon ligand concentration, the predominance of a given complex is also strongly controlled by fluid pH. Under acidic conditions, uranyl complexes most readily with $Cl^-$ and $SO_4^{2-}$[5,16–18], but sea water, many groundwater systems and a number (though not all) of uranium ore deposits are characterized by fluids with near-neutral/slightly alkaline pH ranges[5,16,19–21].

At near-neutral/slightly alkaline pH and under ambient conditions, uranium mobility may be controlled by hydroxyl, biphosphate ($HPO_4^{2-}$), and carbonate ($CO_3^{2-}$) complexes[5,15,16]. It has been suggested that under such pH conditions and at elevated temperatures uranyl-carbonate complexes in particular could play an important role in uranium transport[5,16,20–27]. However, to date, uranyl carbonate complexation has only been experimentally explored at temperatures up to 70 °C[28], with thermodynamic properties only being derived from room temperature experiments[15]. This means that all inferences and models made for elevated temperatures have to date been based on extrapolations of room temperature data. Recent high-temperature experiments on other uranyl complexes have shown that such extrapolations are seldom accurate, often being off by orders of magnitude[17,18,29]. This casts doubt on the accuracy and relevance of any high-temperature models that explicitly invoke carbonate as a potent transport enabler of uranium.

Considering the ubiquity of carbonate complexation in current uranium transport models, experimental verification of the uranium carrying capacity of high temperature carbonate-bearing fluids is required. Hence, we investigated carbonate's contribution to uranium transport under hydrothermal conditions using a combination of experimental approaches. In-situ spectroscopy experiments using Raman and X-Ray Absorption Spectroscopy (XAS) were conducted to characterize the predominant uranyl complexes present in solution, and autoclave solubility experiments were performed to provide direct insights into the degree to which carbonate enhances the hydrothermal mobility of uranium. Overall, experiments were conducted over a temperature range spanning 100–250 °C—a range relevant to most uranium ore deposits and a few waste repository designs. Altogether, these experiments aimed to determine the stoichiometry and thermodynamic properties of the uranyl complexes responsible for uranium's mobility in near-neutral, carbonate-bearing hydrothermal systems. The data collected demonstrate that the stability of uranyl carbonate complexes decreases dramatically with increasing temperature, suggesting that these species may not mediate hydrothermal transport of uranium after all.

## Results

**Raman spectroscopy.** Our study commenced with a Raman spectroscopy investigation on carbonate-bearing solutions in which appreciable concentrations of uranium were dissolved at ambient conditions (further details may be found in "Methods" section). This technique permitted in situ observation of uranium's bonding behavior with carbonate at each given temperature ($T$) and pressure ($P$) condition. Raman experiments were performed on a solution containing 0.012 m $UO_3$ and 0.1 m $NaHCO_3$ (where m denotes moles of solute per kilogram of water). This carbonate concentration was chosen as an intermediate representative of the concentrations found in groundwater and uranium-bearing hydrothermal systems, which altogether typically span a range from 0.001 to 0.2 m, though higher concentrations have been suggested for some extreme systems[19,23]. Spectra were collected at 25, 50, 98, and 146 °C at water vapor pressure. Spectra were collected both immediately upon reaching the desired temperature and after 16–67 h to provide enough time for equilibrium to be reached. Although experiments at higher temperatures were planned, they were not performed due to the discovery of precipitation of uranium from solution, as will be discussed below. In planning these experiments it was assumed that, whereas the behavior of uranium in carbonate solutions is well characterized at ambient conditions and has been somewhat evaluated at $T < 100$ °C[28], at temperatures above 100 °C its behavior in such systems was most in need of attention and experimental verification. At all temperatures, solubility calculations based on thermodynamic properties presented in the PSI Nagra Database[30] as implemented by GEMS Selektor[31] suggested that all uranium should have remained in solution, predominantly in the form of the uranyl tricarbonate complex, $[UO_2(CO_3)_3]^{4-}$. These predictions were confirmed by Raman spectroscopy at 25, 50, and 98 °C, with spectra corresponding to uranyl tricarbonate (Supplementary Fig. 1). However, contrary to predictions from the theoretical model[32], at 146 °C we observed a decrease of uranyl tricarbonate in solution (Fig. 1). This was coupled with an increase in free carbonate and the precipitation of a solid uranium phase. Methodological limitations precluded in situ characterization of this solid phase, but by using the same thermodynamic model later used for our subsequent solubility experiments (which omits uranyl carbonate complexes—details reported in "Methods" section) we determined this phase was likely $UO_2(OH)_2(s)$, though, if added to the model, $Na_2U_2O_7$ was another possibility. Given that the wide body of prior work discussed above has relied on extrapolations similar to those used in our thermodynamic calculations, these results cast some doubt on the stated capability of carbonate-bearing solutions to carry appreciable concentrations of uranium at temperatures ≥150 °C.

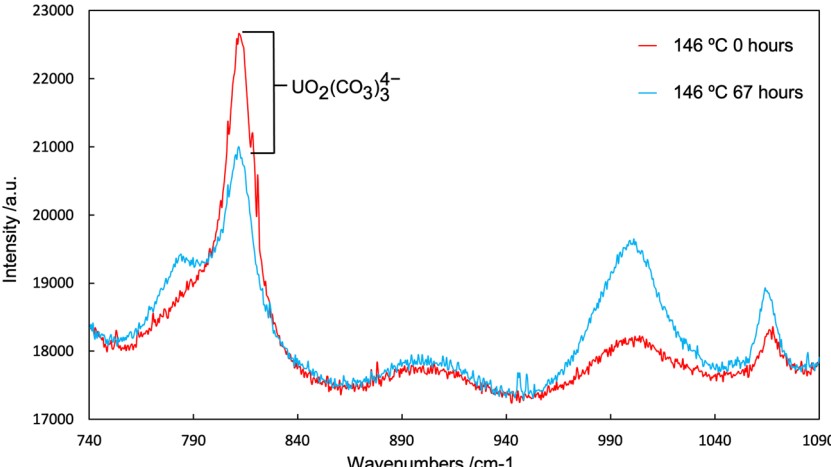

**Fig. 1 Raman Spectra of a solution of 0.012 m UO₃ and 0.1 m NaHCO₃ collected at 146 °C before and after heating for 67 h.** The U–O v1 stretching mode peak associated with the uranyl tricarbonate complex can be seen in both spectra but has diminished after 67 h. The broad peak visible in all spectra at ~792 cm⁻¹ corresponds to a water background signal. Peaks at ~1020 cm⁻¹ and ~1070 cm⁻¹ correspond to HCO₃⁻/CO₃²⁻ and indicate that carbonate was stable in solution. Note the increase in available carbonate after 67 h is consistent with the breakdown of uranyl tricarbonate complexes.

**X-ray absorption spectroscopy (XAS).** To verify the discrepancy between the predicted[30–32] and observed behavior of uranium in carbonate solutions, we further investigated the molecular level structure of the uranyl species present in the studied aqueous solution by XAS using a Hydrothermal Diamond Anvil Cell (HDAC)[33,34] (see Supplementary Methods for details). As with the Raman spectroscopy investigation discussed above, the HDAC XAS technique also permitted in situ solution characterization at temperatures and pressures of interest. This experiment was conducted on a solution containing 0.25 m NaHCO₃ and 0.05 m UO₃. EXAFS data Fourier transforms are reported in Fig. 2 and Supplementary Fig. 2. Spectra were collected up to 125 °C, and similar to the Raman results, they showed a decrease in the spectral features associated with uranyl carbonate complexes with increasing temperature. The XAS data strongly suggest that carbonate complexes of uranium, being predominant at lower temperatures (which can be clearly seen from the U-CO₃ feature on the spectra collected at $T < 100$ °C), become less prevalent when temperature exceeds 100–120 °C, leading to a decrease in the ability of carbonate-bearing solutions to transport uranium.

**Solubility experiments.** Such unexpected and drastic changes in the behavior of uranium in high temperature carbonate-bearing solutions represent a major finding, which significantly alters our understanding of the behavior of uranium in hydrothermal systems. Unfortunately, such behavior precluded any further spectroscopic experiments at higher temperatures as the quantities of uranium that remained in solution were below the detection limits of both in situ Raman and XAS techniques. Thus, to perform a cross-check of these spectroscopic observations, and to determine whether this unexpected behavior is present at temperatures above 150 °C, autoclave solubility experiments were also conducted. Solubility experiments involve the determination of the solubility of solid phases in solutions of interest (see "Methods" section; Supplementary Figs. 3–5). In the case of our experiments, we investigated the ability of carbonate-bearing solutions (up to 0.8 m) to dissolve uranyl hydroxide (UO₂(OH)₂(s)). This phase was determined to be stable at the investigated P-T conditions and solution compositions by both thermodynamic modeling (model details and data sources may be found in the "Methods" section) and post-experiment X-ray diffraction (XRD) measurements of the solids used. This

technique takes advantage of being able to measure the change in U solubility as a function of ligand concentration—in this case carbonate (CO₃²⁻). For example, if uranium in solution were predominantly present as [UO₂(CO₃)₃]⁴⁻, the primary equation describing its solubility would be UO₂(OH)₂(s) + 3CO₃²⁻ ⇌ [UO₂(CO₃)₃]⁴⁻ + 2OH⁻. Based on the associated equilibrium constant, the uranium concentration will increase by three orders of magnitude if carbonate concentration increases by one order of magnitude (i.e., slope of 3 on a log activity of uranium complex vs. log activity of carbonate ion plot). By measuring this relationship, these experiments would determine if UO₂(OH)₂ solubility was in any way correlated to the presence of carbonate in high-T solutions, as well as permit the evaluation of the stoichiometry and thermodynamic molal properties of the predominant uranyl complex controlling solubility. Results from these experiments, collected at 200 and 250 °C and saturated water-vapor pressure, are summarized in Fig. 3 and are reported alongside predicted values of uranium solubility under the same conditions using data from the PSI Nagra Thermodynamic Database as implemented by GEMS Selektor[30,31]. Numerical values can be found in the electronic supplementary data file (Supplementary Tables 1–3 and Supplementary Data 1). It should be noted that the calculated solubilities we report merely present one potential result of room temperature extrapolations. A similar attempt was made by Bastrakov et al.[16], however, the formation constants derived in their work suggested an even greater stability of uranyl carbonate complexes at high temperature—as such, relative to their results, our calculations show a potential minimum in the degree of inaccuracy one might expect using current room temperature data and extrapolation techniques.

In stark contrast to theoretical predications, our data suggest that carbonate has no systematic effect on enhancing the solubility of uranium at elevated temperatures, and that the total solubility of uranium in carbonate-bearing solutions is significantly lower than expected—indeed, discrepancies of up to four orders of magnitude are observed between theory and our experiments. A potential explanation for the observed lack of solubility dependence on carbonate is the formation of a uranyl carbonate solid at elevated temperatures. We deem this unlikely as the separation of solution and solid during autoclave quenching (see Fig. 4 in the supplementary Methods) would likely have preserved any newly formed uranyl carbonate solids,

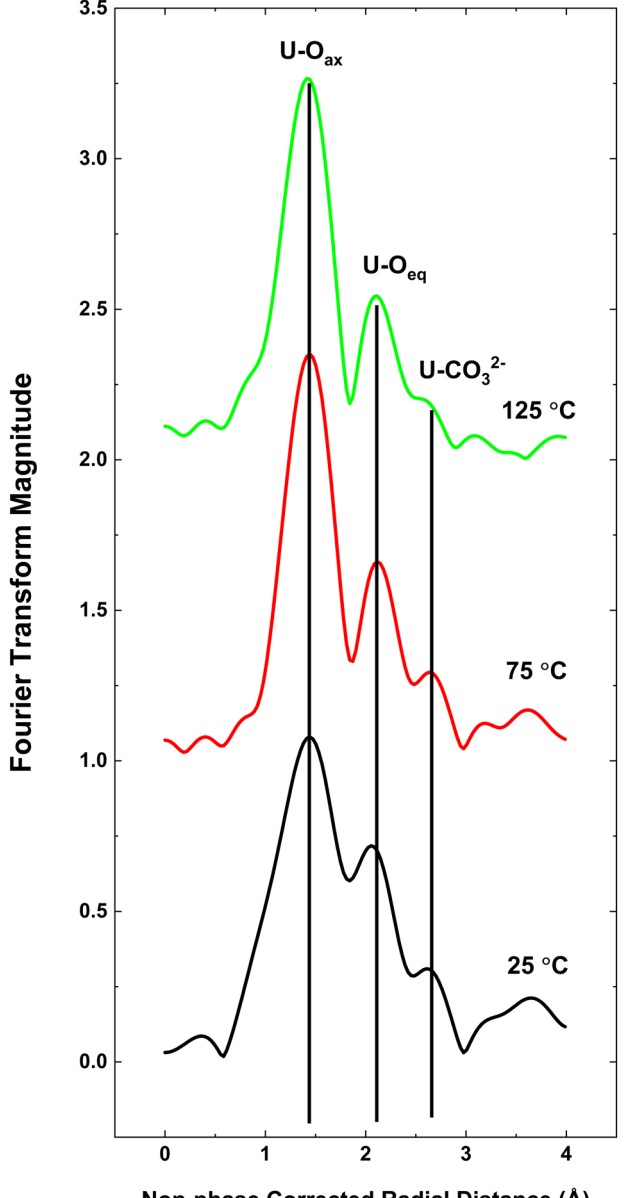

**Fig. 2 Fourier transforms of EXAFS data collected for 0.25 m NaHCO₃ 0.05 m UO₃ solution at 25, 75, and 125 °C.** Note the diminishment, with increasing temperature, of the peak associated with the uranium-C single scattering path at ~2.6 Å and the peaks between 3 and 4 Å typically attributed to multiple scattering paths associated with the carbonate anion.

even if they were highly soluble at room temperature. XRD analysis of the solids used failed to identify any uranyl carbonate phases. Thus, these results confirm the observations made using Raman and XAS techniques, altogether suggesting that uranyl-carbonate complexes do not have any detectable contribution to the mass balance of dissolved uranium at the investigated experimental conditions. By themselves, the Raman and XAS observations could be explained by retrograde solubility of $UO_2(OH)_2$, however given the solubility experiment results we deem this cause is unlikely or at least only a minor contributor to the loss of uranium from solution. It should be emphasized that the experimental conditions (both with regards to carbonate concentrations and temperature) investigated are indeed relevant to a number of natural and engineered (e.g., nuclear waste repository associated) uranium-bearing hydrothermal systems.

The findings reported above cast serious doubt on any models of high-temperature aqueous uranium movement that invoke carbonate as a transporting ligand. Indeed, such models are likely overestimating the mobility of uranium in hydrothermal systems by several orders of magnitude. Our spectroscopic and solubility data suggest that carbonate cannot be invoked as a transporting agent for uranium in ore deposits, nor as a mobility enabler of uranium from nuclear waste in repositories at their peak thermal conditions (at least proximally to the waste itself). Indeed, the carbonate ion should only be considered as a relevant transport enhancer of uranium in natural waters at temperatures below 150 °C.

A major question that arises in the context of our results is, if carbonate is ineffective at enhancing the mobility of uranium at temperatures above 100 °C, what complexes are instead responsible for uranium transport under such conditions? Furthermore, are these complexes more or less stable than theoretically predicted? At room temperature, the next most important group of complexes responsible for uranium transport in near-neutral fluids are the hydroxyl (OH⁻) complexes, and extrapolations suggest that this is also true at elevated temperatures[5,15,16]. However, such extrapolations are solely based on low temperature (25–85 °C) experimental data[35], which, as already illustrated for carbonate, can lead to dubious predictions.

To investigate this alternative, we performed another set of solubility experiments—this time, maintaining a constant carbonate concentration (0.3 m—which based on the results reported in Fig. 3 was presumed to have no effect on uranium solubility) and varying pH. Solution pH was controlled by varying relative ratios of NaHCO₃ and Na₂CO₃, yielding a pH range from ~7 to 10. The results from this new set of experiments are reported in Fig. 4 and the electronic supplementary data file (Supplementary Data 1). Again, we have plotted theoretically calculated values for uranium solubility alongside our experimental results. These calculations assumed no contribution of carbonate speciation to the mass balance of dissolved uranium, with hydroxyl complexes instead being invoked as the primary control on uranium solubility.

When plotted against pH, the uranium concentrations observed in our experiments show a dependence that suggests the predominance of the $UO_2OH^+$ complex consistent with the reaction $[UO_2(OH)_{2(s)} + H^+ \rightleftharpoons UO_2OH^+ + H_2O]$. Furthermore, discrepancies between theoretically calculated uranium solubilities range from about 1–3 orders of magnitude suggesting that the extrapolations of $[UO_2OH^+]$ molal properties from 85 °C[35] are qualitatively valid but require quantitative revision. Revised stability constants for $UO_2OH^+$ and comparisons with data available in the literature are reported in the Electronic Supplement. Hence, our results indicate that the true stabilities of uranyl hydroxyl complexes are significantly higher at elevated temperatures than currently believed.

## Discussion
From the combination of Raman, XAS, and solubility results presented above, it is evident that carbonate complexes are irrelevant at temperatures above 100 °C in geofluids, and that the capability of carbonate to mobilize uranium under hydrothermal conditions has historically been greatly overestimated. This suggests that the invocation of carbonate as a means of transporting uranium in the near-neutral hydrothermal fluids that characterize a number of uranium deposits and transport systems (e.g., the Jáchymov, Czechia, Schwarzwald, Germany; and Hangjinqi, China, deposits)[5,16,20–27] should be treated with skepticism. In addition, in such prior models, the invocation of carbonate as a transporting ligand was not only important for transport of

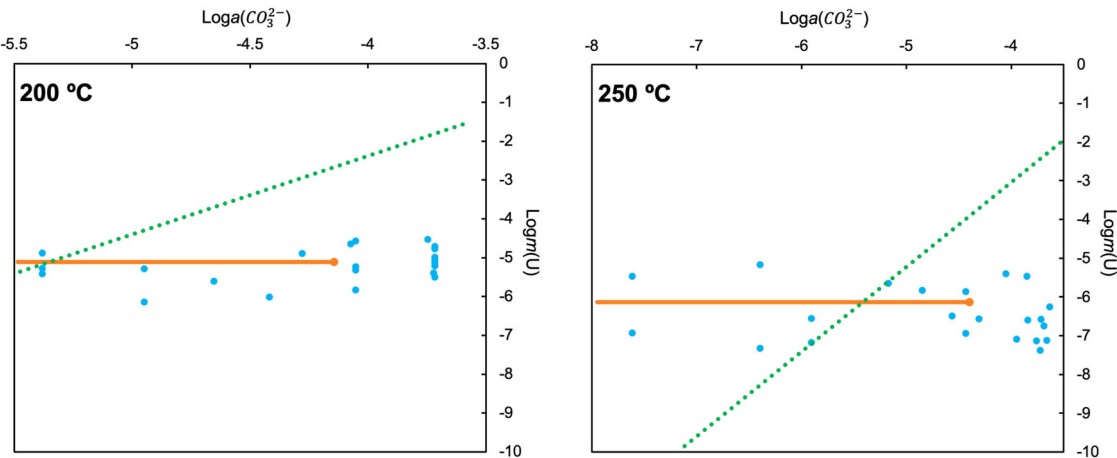

**Fig. 3 Measured and calculated solubilities of UO$_2$(OH)$_2$ as functions of calculated carbonate activity at 200 and 250 °C.** Measured solubilities are shown as dark blue points and calculated solubilities for the same system are shown in light blue. A gradient of 3 for the calculated solubilities would suggest a predominance of the UO$_2$(CO$_3$)$_3^{4-}$ complex, which, however, is not the case as revealed by the experimental results. The dark gray line delineates a range of carbonate activities for groundwater and uranium-bearing hydrothermal systems. Note that data spreads spanning an order of magnitude are typical for autoclave solubility experiments.

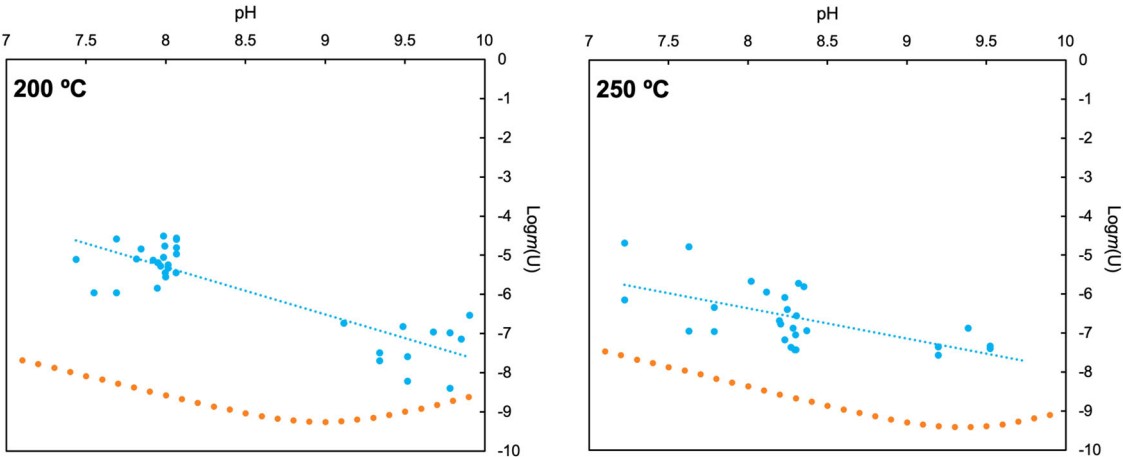

**Fig. 4 Measured and calculated solubilities of UO$_2$(OH)$_2$ plotted as functions of pH at temperature at 200 and 250 °C.** Measured solubilities and trend line are shown in dark blue. Calculated solubilities (assuming no uranyl carbonate speciation) are shown in light blue. Calculated solubilities suggest a predominance of UO$_2$OH$^+$ over most of the pH range shown with a shift towards UO$_2$(OH)$_2$(aq) then UO$_2$(OH)$_3^-$ predominance at pH values > ~9–9.5 as indicated by gradients of 0 and 1 consistent with the reactions [UO$_2$(OH)$_2$(s) ⇌ UO$_2$(OH)$_2$(aq)] and [UO$_2$(OH)$_2$(s) + H$_2$O ⇌ UO$_2$(OH)3$^-$ + H$^+$].

uranium but deposition and accumulation as well. With the supposedly great capability of carbonate bearing fluids to transport uranium, any mechanism that leads to the loss of carbonate from such fluids, e.g., precipitation of solid carbonate phases, was often suggested as a means to induce uranium precipitation and thus lead to the development of economic uranium ore. In the absence of carbonate complexes, these schemes need to be abandoned and some other transport and depositional mechanisms need to be developed. We suggest that in such hydrothermal systems, under near-neutral to alkaline pH conditions, uranium is more likely to be transported as hydroxy complexes, chiefly [UO$_2$OH]$^+$.

Total dissolved uranium in ore-forming fluids varies over a wide range from around $10^{-6}$ to $10^{-2.55}$ m (~240 ppb–670 ppm)[2,4,36–38]. Over the temperature and pH range investigated in our solubility experiments, we observed a maximum recorded uranium solubility of approximately $10^{-5}$ mol/kg, well within the range of dissolved uranium observed in natural systems. Furthermore, some uranium deposits are characterized by fluids whose pHs are controlled by mineral buffers such as Quartz-K-

Feldspar-Muscovite and Kaolinite-Illite or the precipitation of hydroxyl-consuming phases[5,39]. Such buffers and processes tend to produce slightly acidic (pH 4–6.5) fluids, which according to calculations using our newly proposed thermodynamic properties could lead to the transport of significantly higher concentrations of uranium on the order of $10^{-3}$ m (Note how in the equation [UO$_2$(OH)$_{2(s)}$ + H$^+$ ⇌ UO$_2$OH$^+$ + H$_2$O] an increase in the activity of H$^+$, i.e., a decrease in pH, leads to more solubilised uranium). Such concentrations and pH conditions are typically associated with world-class uranium ore deposits such as those in the Athabasca and McArthur basins[2,4,6]. A notable implication of this is that the recent invocations of chloride and sulphate complexes[2,17] as means to mobilise the significant concentrations of uranium required for world-class ore deposits may in fact be unnecessary. Instead, all uranium transport in such deposits could potentially be attributed to uranyl hydroxyl complex formation rather than the actions of other common inorganic ligands. This highlights an important avenue of further pursuit, namely, detailed characterization of the pH dependence of uranium aqueous solubility at elevated temperatures, which prior to

this work has only been investigated up to 85 °C[35], this would permit an accurate evaluation of the relative stabilities of uranyl chloride, sulphate, and hydroxy complexes over a range of geologically relevant pH conditions and ligand concentrations.

The results of this work not only have implications for the study of uranium ore deposit formation, but they have environmental implications as well, specifically with regards to the design of nuclear waste repositories. As exemplified by the recent work of Ewing[22], uranyl-carbonate complexation has often been cited as a primary concern for waste repository design largely due to its ubiquity and potency at room temperature conditions[22,40,41]. At thermal maximum, temperatures within the vicinity of stored high-level radioactive waste may span from 100 to 250 °C[8,10–12,42–44] though naturally this is dependent on repository design. At such temperatures, uranyl-carbonate complexation is suppressed and our results additionally indicate that $UO_2OH^+$ becomes less stable with increasing temperature, leading to the rather counterintuitive conclusion that high temperatures may in fact stifle uranium liberation and transport from nuclear waste and that the risk of liberation increases as the waste cools and uranyl-carbonate complexes become relevant. Naturally we do not suggest that all repositories should be maintained at elevated temperatures but this conclusion suggests that, in terms of carbonate-bearing solutions, greater care should be considered to the later, cooler stages of a repository's life when uranium transport by carbonate-bearing fluids becomes enhanced.

## Conclusions

To conclude, results from the experiments reported here suggest that at temperatures above 150 °C the uranium carrying capacity of carbonate-bearing fluids typical of many uranium ore deposits and of concern for radioactive waste repositories may have been historically overstated. We have demonstrated that at such conditions carbonate is incapable of enhancing uranium mobility, which suggests that a significant number of uranium transport models may require revisiting and revision. On the other hand, we have also found that the carrying capacity of uranyl hydroxy complexes is significantly higher than historically believed, and that they could potentially account for a significant portion of the uranium carrying capacity of a wide range of geofluids.

## Methods

**Raman spectroscopy**. Raman spectra were collected using a Horiba Jobin Yvon Evolution Raman spectrometer. The spectrometer is equipped with an 800 mm focal length, a polarized 532 nm, 250 mW Nd:YAG laser, an edge filter with a Stokes edge of 50 cm$^{-1}$, a 1024 × 256 pixel CCD detector, an 1800 line/mm grating, and a confocal Olympus microprobe with an adjustable slit entry set to 200 μm. All spectra were obtained through a 20× objective lens (SLMPLN, Olympus) using light that was backscattered from the sample.

Expanding on the procedures first developed by Chou et al.[45] the samples were contained in fused silica capillary tubing (Technical Glass Inc., inner and outer diameters of 1 mm and 2 mm, respectively), which were sealed by a hydrogen-oxygen torch. The capillary tubes were only half filled with liquid sample prior to sealing, and pressure was controlled through equilibrium with the gas phase (i.e., at the saturation vapor pressure of the sample). Temperature control was provided by a Linkam THMS600 stage (Linkam Scientific Instruments) coupled to a T96 temperature controller. In order to provide improved temperature control to the sample, the capillary tubing was housed in a custom-made aluminum heating block, manufactured to sit directly on top of the silver heating stage of the THMS600. This aluminum block is 16 mm in diameter, 20 mm in height, and has a 1 mm channel drilled through it to house the fused silica capillary tubing. Temperature calibration was performed using a K-type thermocouple cemented in a 1 × 2 mm capillary tube containing air.

Samples were prepared by dissolving a measured amount of $UO_3$ in a 0.1 m $NaHCO_3$ solution made using degassed de-ionized water. Separate samples of 0.012 m U + 0.1 m $NaHCO_3$ were each heated at 50, 98, and 146 °C for 16–65 h, with Raman spectra taken at those temperatures, both pre-heating and post-heating. The inner surface was also investigated by a camera attachment to our Raman system to observe any possible precipitate.

**XAS experiments**. We performed X-ray absorption spectroscopy (XAS) at the uranium (U) $L_{III}$-edge (17166.3 eV) at beamline 11-2 at the Stanford Synchrotron Radiation Lightsource (SSRL). We performed these measurements up to 125 °C and 350 MPa using a hydrothermal diamond anvil cell (HDAC) with specially designed radioactive enclosure on a uranyl ($UO_2^{2+}$) carbonate ($CO_3^{2-}$) solution with $[UO_2^{2+}]$ = 0.05 m and $[CO_3^{2-}]$ = 0.25 m. The sample solution was prepared from a 2 mL solution of $UO_3 \cdot H_2O$ dissolved in $HClO_4$ ($[U^{6+}]$ = 0.1 m, $[HClO_4]$ = 0.5 m) by adding 1.2 mL of 0.5 m NaOH and 4.2 mL of 0.25 m $NaHCO_3$. The measured sample consists of an aqueous liquid plus vapor bubbles that is placed into the HDAC sample chamber, defined by a 700 μm hole drilled at the center of a 125 μm (100 μm when compressed) thick rhenium (Re) gasket with an outer diameter of 3000 μm. Heating was achieved by resistive heaters near the diamonds, and temperature was measured with K-type thermocouples attached to each diamond. Pressure was achieved through application of force by tightening screws on the HDAC and, the pressure was estimated based on solution density. Further details about the experimental set-up are described by Dhakal et al.[33,34].

XAS data acquisition in fluorescence mode was made using a 100-element Canberra Ge solid-state monolith detector placed in the standard 90° orientation to the incident X-ray beam. The beam was focused to 250 μm in the horizontal direction and 1 mm in the vertical directions using Kirkpatrick-Baez mirrors, and the incident photon energy of the beam was varied using a double-crystal Si (220) monochromator. Detuning of the monochromator crystal was set at 15%. A yttrium (Y) foil in the beam path allowed energy calibration and calibration of XAS spectra to the Y K-edge (17,038 eV). The 1st derivative with respect to energy of each XAS spectra was calibrated to this edge and data reduction including fitting and subtracting a background function using a pre-edge and post-edge function was performed using the Athena software package[46].

**Solubility experiments**. Solubility experiments were designed to permit the evaluation of both the stoichiometry and formation constants of the predominant uranyl complex responsible for stabilizing uranium in high-temperature carbonate-bearing solutions. Experiments were conducted at 200 and 250 °C at saturated water-vapor pressure (SWP) using $UO_3$, which was converted to the reference solid ($UO_2(OH)_2$) through exposure to experimental solutions and conditions as a reference solid. The experiments that investigated carbonate complexation (Series 1), were set up to have the reference solid interact with solutions containing variable quantities of carbonate added as $NaHCO_3$. In the experiments investigating hydroxyl complexation (Series 2) these solutions instead contained a constant total carbonate concentration, which was added as varying ratios of $NaHCO_3$ and $Na_2CO_3$ which was used to vary pH. Additionally, to ensure that our chosen activity model was applicable, all solutions also contained 1–2 m NaCl. Uranium is a redox sensitive element with its hexavalent and tetravalent oxidation states being the most common in natural systems. The speciation behaviors of both valence states are significantly different[15,18,47], thus, to prevent interference and ensure that experiments were only characterizing the solubility and speciation of the hexavalent state a 1:1 $Mn_2O_3/Mn_3O_4$ mix oxygen fugacity buffer was introduced, which ensured that conditions inside each autoclave remained sufficiently oxidizing to stabilize U(VI) and prevented any U(IV) formation.

In Series 1 experiments, carbonate concentration was varied using $NaHCO_3$ (Acros Organics, ACS 99.7%) over a range of 0.001–0.4 m, with care being taken to ensure an ionic predominance of NaCl (Fisher Chemical, Certified ACS), whose concentration was kept at either 1 or 2 m. An ionic predominance of NaCl was required in order to permit the usage of the modified extended Debye-Hückel model[48–50]—activity model details are discussed further below. Based on recent high-temperature experimental data[18] thermodynamic calculations suggested that such concentrations of NaCl result in negligible uranyl-chloride species formation at the pH conditions and temperatures investigated. A few solutions were made with higher $NaHCO_3$ concentrations (up to 0.8 m) but their behavior differed little from those with lower concentrations—thus, to avoid any precipitation and minimize activity model issues, $NaHCO_3$ concentrations were generally kept below 0.4 m. Based on the models for $CO_2$ solubility in water and NaCl predominated fluids reported by Duan and Sun[51], such concentrations of carbonate were stable in solution with outgassing of $CO_2$ deemed unlikely. In Series 2 experiments the concentration of carbonate was kept constant at 0.3 m, where, based on Series 1 experiments, carbonate complexation was presumed negligible. This carbonate was added as varying ratios of $NaHCO_3$ and $Na_2CO_3$ (Fisher Chemical, Certified ACS, Anhydrous) and was used to both control and buffer pH. Being able to vary pH in such a stable manner allowed us to investigate the relation between pH and uranium solubility, thus permitting the investigation of uranyl hydroxyl speciation. Solution compositions may be found in the Supplementary Material.

Experiments were conducted using Teflon-lined titanium (Commercial Grade 2) autoclaves into which carbonate-bearing solutions were placed alongside separate small Teflon holders containing the oxygen fugacity buffer and uranium reference solid. Perforated Teflon plugs were placed in all holders to prevent escape of any particulates, while still permitting interaction between solid reagents and solution/internal atmosphere. At all times, reagents were solely in contact with Teflon surfaces, thus precluding any unexpected chemical interactions with Ti/TiO$_2$. Specific volumes of experimental solution were added such that the holders containing uranium were only submerged at the target experimental temperature via thermal volumetric expansion. The oxygen fugacity buffer holder was made tall

enough such that it was never submerged. Loaded autoclaves were initially heated to 150 °C for 2 days to permit equilibration between the oxygen fugacity buffer, the atmosphere within the autoclave and the uranium solid—this ensured that subsequent solution–solid interactions solely involved hexavalent uranium. The experimental solution did not interact with the uranium solid during this pre-heating phase. After this preheating phase, solutions were heated to the target temperature and maintained at such for 5 days (See the Supplementary Material for details on how this length of time was chosen) to ensure complete equilibration between uranium solids and carbonate solutions. Autoclaves were then extracted and quenched in air to isolate solutions from solids. After opening, both holders were removed and concentrated nitric acid (MilliporeSigma AqueousOmni*Trace*) was added to the experimental solutions were then allowed to soak for 24 h, this was done to dissolve any uranium that may have precipitated during quenching. Solutions were then extracted from autoclaves and uranium concentrations were measured using Inductively Coupled Plasma Mass Spectrometry (ICP-MS).

To accurately determine formation constants a stable reference solid was required. Thermodynamic calculations suggested Paulscherrerite ($UO_2(OH)_2$) was stable under the chosen experimental conditions. To confirm this, samples of the reference solid were extracted after the experiments were quenched and were characterized using powder X-Ray Diffraction (PXRD). Quantitative phase analysis was performed using the Rietveld method[52] and confirmed the sole presence of $UO_2(OH)_2$, primarily in its alpha form (~66%) with minor quantities (~33% total) of its beta and gamma polymorphs. Measurement results may be found in the electronic supplement data file.

**Thermodynamic calculations.** The calculation of solution pH values at temperature and formation constants from experimental data required the activity of all species in solution to be calculated, which required a suitable activity model. While there are many such activity models available for ambient conditions, comparatively few are applicable to solutions with appreciable ionic strengths at elevated temperatures. One of the most reliable is the Extended Debye–Hückel equation of state modified for solutions dominated by 1:1 electrolytes[48–50] (e.g., HCl, NaCl, and NaOH) (Eq. (1)).

$$\log \gamma_i = -\frac{A \cdot [Z_i]^2 \cdot \sqrt{I}}{1 + B \cdot \mathring{a} \cdot \sqrt{I}} + \Gamma + b_\gamma I, \qquad (1)$$

where $A$ and $B$ are the Debye–Hückel parameters, $\gamma_i$, $Z_i$, $\Gamma$, and $\mathring{a}$ are the individual molal activity coefficients, the ionic charge, a molarity to molality conversion factor and ionic size of ion "$i$". The effective ionic strength calculated using the molal scale is $I$ and $b_\gamma$ is the extended-term parameter for the chosen 1:1 background electrolyte. This necessity for a 1:1 dominant background electrolyte is why all experimental solutions contained 1–2 molal NaCl. Note that Eq. (1) was only used to calculate the activity coefficients of charged species, whereas the activities of neutral species were calculated using a simplified form of Eq. (1) described in Eq. (2).

$$\log \gamma_i = \Gamma + b_\gamma I \qquad (2)$$

In all thermodynamic calculations, we defined the experimental system with the following aqueous species: $H^+$, $OH^-$, $Cl^-$, $HCl^0$, $NaCO_3^-$, $NaHCO_3^0$, $Na^+$, $NaCl^0$, $NaOH^0$, $CO_2^0$, $CO_3^{2-}$, $HCO_3^-$, $UO_2^{2+}$, $UO_2(OH)_{2(cr)}$, $UO_2Cl^+$, $UO_2Cl_2^0$ using data sourced from Shock et al.[53], Sverjensky et al.[54], Tagirov et al.[55], Miron et al.[56], and Migdisov et al.[18]. Data for $H_2O$ were sourced from the work of Marshall and Franck[57].

## Data availability
All data generated or analysed in this study have been reported in the Supplementary Data 1

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

## Acknowledgements

The authors would like to thank Rahul Ram, Noah van Hartesveldt, Joshua Taylor White, Chelsea Wren Neil, Oana Marina, and Doug Ware for the invaluable technical assistance they provided during the course of this work. This research was supported by the Department of Energy's Spent Fuel and Waste Disposition campaign and the Laboratory Directed Research and Development program of Los Alamos National Laboratory under project number 20180007DR; the ARC Research Hub on Australian Copper-Uranium (project number: IH130200033), funded by the Australian Research Council, BHP Olympic Dam, OZ Minerals and the South Australian Department of State Development; and an Australian Government Research Training Program Scholarship to A. Kalintsev. Los Alamos National Laboratory, an affirmative action and equal opportunity employer, is managed by Triad National Security, LLC, for the National Nuclear Security Administration of the U.S. Department of Energy under contract 89233218CNA000001. Use of the Stanford Synchrotron Radiation Lightsource, SLAC National Accelerator Laboratory, was supported by the U.S. Department of Energy, Office of Science, Office of Basic Energy Sciences under Contract No. DE-AC02-76SF00515.

## Author contributions

A.M. conceived the research. A.K. conducted the solubility experiments, generated the solubility model and assembled the manuscript, A.M. and J.Br. generated the solubility model and participated in the manuscript's writing, C.A., conducted solubility and Raman experiments and wrote the relevant methods section, H.X. wrote a proposal to obtain synchrotron XAS beamtime, J.Ba., R.A.M., N.A., and X.G. conducted XAS HDAC experiments, and all wrote the relevant methods section. H.X., F.C., H.V., C.J-C., Y.W., E.M., H.B., and R.R. participated in discussions and assisted in writing the manuscript.

## Competing interests

The authors declare no competing interests.
