## [Peer Review File · Communications Chemistry]

Reviewers' comments:

Reviewer #1 (Remarks to the Author):

The present manuscript provides experimental evidence that carbonate ligand is not relevant for transporting uranium at temperatures higher than 100 C and therefore "can no longer be considered key to the formation of uranium ore deposits or as an enabler of uranium transport from nuclear waste repositories at elevated temperatures."

A major issue with the manuscript is that the authors base their selling pitch "loss of a titan" on their suggestion that previous studies somehow considered carbonate species as a "key" in uranium transport.

While carbonate and bicarbonate species were considered to play a role at specific conditions, the previous models of uranium transport and deposition considered other ligands like sulfates, phosphates, chlorides, and hydroxides as key players. See Skirrow et al. (2009) – This is how Skirrow et al. mention carbonate and bicarbonate species "However, uranium transport may be dominated by bicarbonate or oxyhydroxide or sulfate complexes where aqueous phosphate concentrations are low relative to other ligands (e.g., due to elevated Ca²⁺)."

There is merit to the experimental outcome of the research, the authors provide first experimental evidence that carbonate and bicarbonate uranyl species are not forming at hydrothermal conditions. But trying to frame this as disproving conclusions of previous studies and models on uranium transport is farfetched. Or at least it definitely needs some strong arguments and discussion. There might even be studies out there that claimed that carbonate and bicarbonate ligands play a key role (at ambient conditions they can play a role), but which studies concerning elevated temperatures, base their main conclusions on this? There is definitely a need of experimental data on uranium speciation at elevated temperatures and a lot of existing data based on extrapolations lead to wrong results!

lines 35-40. This issue is not relevant or is "little" relevant for nuclear waste repositories. The temperatures are unlikely to reach 200 C, maybe at the contact with the waste canister and for a short period of time, with temperatures much lower expected closer to the natural seal rock (Zheng et al. 2015). In fact, based on the waste content, the temperatures are not expected to exceed 100 C at the canister contact (Finsterle et al. 2019). Generally, temperatures related to waste repositories are expected to remain below 100-80 C and the carbonate uranium species can be relevant for the transport of uranium in these systems (they form around ambient temperatures). This is why they are mentioned in relation to waste repositories and not because they are necessarily considered to be major species > 100 C. This was not clearly known, was maybe speculated, but now the authors provide experimental data showing that this is not the case.

These extreme temperatures that the authors cite are maybe relevant for low thermal conductivity environments like waste repositories in salt but in this case high salinity models are relevant. The authors should provide clear evidence for waste repository designs where such high temperatures are expected in the geological environment of the repository.

"potentially higher temperatures being reached at the surface of stored fuel pellets³" – this is incorrect. In the cited paper, the authors refer to the temperature of the pellets during reactor operation and not during storage. Normally radioactive waste is largely cooled down during predisposal activities.

Ewing (2015) "The distribution of elements is not homogeneous within a single pellet (Fig. 1) because of the steep thermal gradients that exist (as high as 1,700 °C at the centre of the pellet and decreasing to 400 °C at its rim). Thermal excursions during reactor operation can additionally cause a coarsening of the grain size, as well as extensive microfracturing."

"At near-neutral/slightly alkaline pH and ambient conditions, uranium mobility is controlled by hydroxyl and particularly carbonate (CO₃²⁻) complexes, and conventional wisdom suggests that this holds true at elevated temperatures as well^{3,5,8}." – This statement is incomplete/deceiving.

Studies suggest that carbonate complexes may play a role, but with no definitive statement. In fact, uranyl phosphate, sulfate, fluoride, chloride, and hydroxide complexes are believed to play a role in uranium transport at hydrothermal conditions. See "2.3 URANIUM TRANSPORT - SUMMARY" from Skirrow et al. (2009) whom the authors cite. The introduction of the manuscript reads a bit like all previous studies concluded that the transport of uranium was due to the carbonate ligand. This is not the case, yes carbonate ligand was mentioned as a possible contender but Skirrow et al. (2009) mentions other complexes to play a much greater role. The authors need to clarify what sits behind "conventional wisdom".

"This casts doubt on the accuracy and relevance of any high temperature models that invoke carbonate as a potent transport enabler of uranium" – Again this statement sounds like all previous models concluded that carbonate was the only ligand responsible for uranium transport while these models generally invoke other ligands and only in some specific conditions, they mention that carbonate and bicarbonate complexes may have played a role.

"experiments were conducted over a temperature range spanning 100-250 °C – a range relevant to most uranium ore deposits and a number of waste repository designs." – again this statement is extreme when it comes to waste repository designs which mostly consider temperatures < 100 C.

The "PSI Nagra Thermodynamic Database" Thoenen et al. (2014) is meant to be used at ambient conditions, although it contains data such as enthalpies and heat capacities for some components. The version implemented in GEM-Selektor "provided a set of MHKF parameters for numerous aqueous uranium complexes" – Bastrakov et al. (2009).

Bastrakov et al. (2010) note very large discrepancies for the temperature dependence when compared with the NEA reviews. So, it is not surprising that using this dataset for calculations at elevated temperatures, the authors obtained strange results. It is already known that most HKF data derived from correlations and extrapolations for lanthanides and actinides lead to wrong results.

Bastrakov et al. (2010) write: "However, the comparison of these data with data from the NEA shows dramatic differences in the predicted stabilities of uranium complexes (Figure 3.1; stability of UCl_3 and $UO_2CO_3(aq)$). In a note to their report, Thoenen and Kulik (2003) stressed that their dataset contains numerous thermodynamic data that were taken from the literature without being critically reviewed. Thus we preferred to omit their dataset from our compilation."

A solution to this would be to try to get the hands on the dataset of Bastrakov et al. (2010) or derive RBM parameters from the few NEA points at elevated temperatures, like Bastrakov et al. (2010) did. This might still result in wrong extrapolations at elevated temperatures. (see their Figure 3.1 for $UO_2CO_3(aq)$ with 3-4 log units discrepancies at $T > 200$ C for PSI+HKF vs NEA RBM model).

Reviewer #2 (Remarks to the Author):

I am glad to review the paper by Alexander Kalintsev and coauthors. The authors challenge the previous view – carbonate is an important ligand responsible for hydrothermal transport of uranium in alkali or near neutral fluids, based on their in situ spectroscopic and ex situ solubility experimental observations. As the authors have suggested in the paper, the hydrothermal transport of uranium is important for either uranium exploration or nuclear waste storage. The topic of this study is of particular significance and broad interest, no matter how human beings are going with uranium resources. Therefore, I suggest the paper to be published in "Communications Chemistry" after suitable revisions.

Comments of this paper are given as following.

1. The authors inferred that $UO_2(CO_3)_3^{4-}$ is unstable at elevated temperatures as the signal of $UO_2(CO_3)_3^{4-}$ disappeared at ≥ 146 °C. I noticed that the signal of CO_3^{2-} and HCO_3^- also disappeared at 146 °C. Then, where did carbonate go in the fluids at ≥ 146 °C? Was it decomposed or precipitated as solids? And, can the fluid still be termed as a carbonate-bearing solution (line 120 in the manuscript)?

2. The authors observed a solid precipitate (termed as 'uranium phase' in line 92 of the manuscript) after the sample was heated at 146 °C for 16 h, but didn't provide any further characterization. Please add some photographs of the solid precipitate in the supplementary material. Also, I suggest the authors to conduct some further scanning electron microscopic (SEM) observations and characterize its composition with energy diffraction spectrometer (EDS) or if possible, conduct some micro-XRD analysis. After conducting further characterization of the solid precipitate, the involvement of carbonate, silica, or uranyl can be better constrained.
3. Did you observe any precipitates in your XAS experiments?
4. The authors have conducted some thermodynamic calculations to derive the pH and some other properties of the fluid. But some calculation results are not listed in the experimental data. Please provide more details of the thermodynamic calculations (including composition-pressure-temperature conditions, species distributions, pH, activities coefficients and etc.) in the files.
5. In the solubility experiments, the author viewed the total amount of CO₃²⁻ (sourcing from NaHCO₃) as dissolved amount of carbonate. Can carbonate precipitates form in the system during heating? In addition, I noticed that the log_a[CO₃²⁻] is much lower than log_m[CO₃²⁻]. Does that mean the activity coefficient is very low at such conditions (below 10⁻³)? Please list more detailed data of the thermodynamic calculations (also in question 4).
6. There are some format errors in the manuscript and supplementary material. Some sources didn't show in the supplementary material (line 137, line 239, line 249, line 254, line 257 in the supplementary material). The multiplicative sign '20x' should be used in front of the objective lens instead of '20X' (line 10, in the supplementary material). A space is needed between '16' and 'h' to represent 16 hours (lines 24, 29, 36, and 40 in the supplementary material). '-' and '–' should be clearly and consistently used in the main text, references, and figures! Please correct!
7. The figures need further optimization. For example, the font of temperature marked in figures 3 and 4 are too larger than the tick labels and the title of the x/y axis.
8. Both fluid temperature and pressure should be specified in the main text.
9. Lin 75, (m denotes moles of solute ...)
10. For HDAC experiments, how to determine the composition of the loaded fluid sample. In my opinion, the ultimate composition is different from that in the bulk solution. During sample loading, evaporation will result in an increase in the fluid concentration. In addition, a fluid pressure of 350 MPa is much higher than the geological counterparts.

Reviewer #3 (Remarks to the Author):

COMMSCHEM-21-0017-T Loss of a Titan: Uranium carbonate complexes at elevated temperatures

Authors: Alexander Kalintsev, Artas Migdisov*, Christopher Alcorn, Jason Baker, Joel Brugger, Robert A. Mayanovic, Nadib Akram, Hongwu Xu, Hakim Boukhalfa, F. Caporuscio, H. Viswanathan, C., Jove-Colon, Y. Wang, E. Matteo, Robert Roback

Assessment

This paper is a systematic experimental study, employing an impressive variety of complementary techniques, of the effect of carbonate complexes on the solubility of uranyl hydroxide, (UO₂(OH)₂(s)), in sodium carbonate and bicarbonate solutions as the temperature is raised from ambient to hydrothermal conditions (250°C, psat). Raman spectroscopy and XAS using hydrothermal diamond anvil cells, composition-dependent solubility studies, and chemical equilibrium modelling are used to demonstrate that the uranyl ion, UO₂(2+) does not form stable carbonate complexes at these temperatures and that its solubility is governed by hydrolysis reactions under these experimental conditions. This finding has significant implications for some types of geological repositories for nuclear waste management and hydrothermal ore body formation. The experiments are very well designed and appear to have been carefully carried out. The discussion and conclusions are appropriate, and the topic is noteworthy.

Recommendation:

The paper is extremely well written and makes an important contribution to the literature. It should be published, with very minor editorial revisions.

Suggested Changes/Corrections

The paper is well written, and I have only few specific suggestions:

- The paper is somewhat verbose for a Chemical Communication. The text could be more succinct.
- The captions to the figures need to be more carefully worded. Figures 3 and 4 should specify the mineral whose solubility is plotted.
- Lines 125 to 130. Was the equilibrium solid phase shown to be uranyl hydroxide, $\text{UO}_2(\text{OH})_2(\text{s})$, or could it have transformed into uraninite, $\text{UO}_2(\text{s})$, or a uranyl carbonate mineral?
- Line 136 and elsewhere: The term "uranium solubility" is imprecise. The saturating solid phase should be identified.

Reviewer 1

A major issue with the manuscript is that the authors base their selling pitch "loss of a titan" on their suggestion that previous studies somehow considered carbonate species as a "key" in uranium transport.

While carbonate and bicarbonate species were considered to play a role at specific conditions, the previous models of uranium transport and deposition considered other ligands like sulfates, phosphates, chlorides, and hydroxides as key players. See Skirrow et al. (2009) – This is how Skirrow et al. mention carbonate and bicarbonate species "However, uranium transport may be dominated by bicarbonate or oxyhydroxide or sulfate complexes where aqueous phosphate concentrations are low relative to other ligands (e.g., due to elevated Ca²⁺)."

It was never the intention of the paper to imply that uranyl-carbonate complexes have been invoked in **all** hydrothermal uranium transport systems and the reviewer's interpretation was likely a result of poor wording on our account. In order to clarify our standpoint on this matter the section that discusses their importance (lines 50-70) has been rewritten to more clearly state that their importance is closely linked to fluid pH and that they do not necessarily predominate in all systems.

There is merit to the experimental outcome of the research, the authors provide first experimental evidence that carbonate and bicarbonate uranyl species are not forming at hydrothermal conditions. But trying to frame this as disproving conclusions of previous studies and models on uranium transport is farfetched. Or at least it definitely needs some strong arguments and discussion. There might even be studies out there that claimed that carbonate and bicarbonate ligands play a key role (at ambient conditions they can play a role), but which studies concerning elevated temperatures, base their main conclusions on this?

Again, we are not claiming that carbonate has been suggested as the primary uranium transport enabler in all hydrothermal systems but that it is important under specific chemical conditions that are somewhat common in uranium-bearing hydrothermal systems, in addition we have cited a large body of work that invokes carbonate as a uranium transport enabler. Relevant citations were rather scattered in the originally submitted version of the manuscript, we have now concentrated them on line 64 for greater visibility (citations 5,16,20-27). All these references cite carbonate complexes as a potential uranium vector in hydrothermal systems.

There is definitely a need of experimental data on uranium speciation at elevated temperatures and a lot of existing data based on extrapolations lead to wrong results!

We absolutely agree!

Lines 35-40. This issue is not relevant or is "little" relevant for nuclear waste repositories. The temperatures are unlikely to reach 200 C, maybe at the contact with the waste canister and for a short period of time, with temperatures much lower expected closer to the natural seal rock (Zheng et al. 2015). In fact, based on the waste content, the temperatures are not expected to exceed 100 C at the canister contact (Finsterle et al. 2019). Generally, temperatures related to waste repositories are expected to remain below 100-80 C and the carbonate uranium species can be relevant for the transport of uranium in these systems (they form around ambient temperatures). This is why they are mentioned in relation to waste repositories and not because they are necessarily considered to be major species > 100 C.

This was not clearly known, was maybe speculated, but now the authors provide experimental data showing that this is not the case.

These extreme temperatures that the authors cite are maybe relevant for low thermal conductivity environments like waste repositories in salt but in this case high salinity models are relevant. The authors should provide clear evidence for waste repository designs where such high temperatures are expected in the geological environment of the repository.

The reviewer comment was addressed in two ways:

- 1) The reviewer is correct in stating that the majority of radioactive waste repository designs are unlikely to reach local rock temperatures >100 °C and we have changed the introduction's wording (Lines 44-48) to make note of this fact. However, studies on the storage of the American dual-purpose canister design concept are an exception to this general fact. It is from these studies that we draw the suggestion of higher repository temperatures – see references 10-12, lines 47-48.
- 2) In the introduction of the original version of our manuscript the waste repository topic was overweighed. However, knowledge of the mechanisms of hydrothermal transport of U is equally important for

understanding of processes occurring during ore formation and for assessing the consequences of nuclear accidents, such as Fukushima Daiichi disaster. In the revised version we tried to present all these topics in a more balanced way (lines 37-50)

“potentially higher temperatures being reached at the surface of stored fuel pellets?” – this is incorrect. In the cited paper, the authors refer to the temperature of the pellets during reactor operation and not during storage. Normally radioactive waste is largely cooled down during predisposal activities.

Ewing (2015) “The distribution of elements is not homogeneous within a single pellet (Fig. 1) because of the steep thermal gradients that exist (as high as 1,700 °C at the centre of the pellet and decreasing to 400 °C at its rim). Thermal excursions during reactor operation can additionally cause a coarsening of the grain size, as well as extensive microfracturing.”

We concede on this point and have removed this line from the text

“At near-neutral/slightly alkaline pH and ambient conditions, uranium mobility is controlled by hydroxyl and particularly carbonate (CO₃²⁻) complexes, and conventional wisdom suggests that this holds true at elevated temperatures as well^{3,5,8}.”

This statement is incomplete/deceiving. Studies suggest that carbonate complexes may play a role, but with no definitive statement. In fact, uranyl phosphate, sulfate, fluoride, chloride, and hydroxide complexes are believed to play a role in uranium transport at hydrothermal conditions. See “2.3 URANIUM TRANSPORT - SUMMARY” from Skirrow et al. (2009) whom the authors cite.

The introduction of the manuscript reads a bit like all previous studies concluded that the transport of uranium was due to the carbonate ligand.

This is not the case, yes carbonate ligand was mentioned as a possible contender but Skirrow et al. (2009) mentions other complexes to play a much greater role. The authors need to clarify what sits behind “conventional wisdom”.

“This casts doubt on the accuracy and relevance of any high temperature models that invoke carbonate as a potent transport enabler of uranium” – Again this statement sounds like all previous models concluded that carbonate was the only ligand responsible for uranium transport while these models generally invoke other ligands and only in some specific conditions, they mention that carbonate and bicarbonate complexes may have played a role.

We believe changes made in the introduction mentioned earlier adequately address this concern. Poor wording on our part evidently seemed to imply that we were asserting that carbonate complexes are important for **all** uranium hydrothermal transport systems which simply is not the case. In addition, the area of the text highlighted by the reviewer (lines 61-64 in the revised manuscript) has also been adjusted to present a more balanced view and make our point clearer.

“experiments were conducted over a temperature range spanning 100-250 °C – a range relevant to most uranium ore deposits and a number of waste repository designs.” – again this statement is extreme when it comes to waste repository designs which mostly consider temperatures < 100 C.

Refer to the alterations we mentioned earlier that highlight the Dual-purpose canister concept

The “PSI Nagra Thermodynamic Database” Thoenen et al. (2014) is meant to be used at ambient conditions, although it contains data such as enthalpies and heat capacities for some components. The version implemented in GEM-Selektor “provided a set of MHKF parameters for numerous aqueous uranium complexes” – Bastrakov et al. (2009).

Bastrakov et al. (2010) note very large discrepancies for the temperature dependence when compared with the NEA reviews. So, it is not surprising that using this dataset for calculations at elevated temperatures, the authors obtained strange results. It is already known that most HKF data derived from correlations and extrapolations for lanthanides and actinides lead to wrong results.

Bastrakov et al. (2010) write: “However, the comparison of these data with data from the NEA shows dramatic differences in the predicted stabilities of uranium complexes (Figure 3.1; stability of UCl₃ and UO₂CO₃ (aq)). In a note to their report, Thoenen and Kulik (2003) stressed that their dataset contains numerous thermodynamic data that were taken from the literature without being critically reviewed. Thus we preferred to omit their dataset from our compilation.”

A solution to this would be to try to get the hands on the dataset of Bastrakov et al. (2010) or derive RBM parameters from the few NEA points at elevated temperatures, like Bastrakov et al. (2010) did. This might still result in wrong extrapolations at elevated temperatures. (see their Figure 3.1 for UO₂CO₃(aq) with 3-4 log units discrepancies at T > 200 C for PSI+HKF vs NEA RBM model).

The reviewer is correct in placing suspicion in the results of the calculations we report based on extrapolation of room temperature data for uranyl carbonate complexes. We report them more for illustrative purposes to show potentially how inaccurate such extrapolations can be not as an objective truth. We compared formation constants reported by the GEMS implementation of uranyl carbonate complexes with those reported in Bastrakov et al. (2010) and found that the values reported in the latter invariably suggest even *greater* stability (by an order of magnitude or more) of uranyl carbonate complexes at temperature. As such we elect to maintain our illustrative calculations as is to show a 'best case' for the degree of inaccuracy one might expect in the case of uranyl carbonate complexes. We have added a brief additional discussion clarifying this point around lines 174-179.

Reviewer 2

1. *The authors inferred that $UO_2(CO_3)_3^{4-}$ is unstable at elevated temperatures as the signal of $UO_2(CO_3)_3^{4-}$ disappeared at ≥ 146 °C. I noticed that the signal of CO_3^{2-} and HCO_3^- also disappeared at 146 °C. Then, where did carbonate go in the fluids at ≥ 146 °C? Was it decomposed or precipitated as solids? And, can the fluid still be termed as a carbonate-bearing solution (line 120 in the manuscript)?*

Well spotted, we did not notice this missing detail. The same solution composition was remeasured at the same temperature. This time we were able to capture evidence of missing soluble uranium alongside the increase of carbonate/bicarbonate in solution consistent with the liberation of carbonate from breaking down uranyl-carbonate complexes. Figure 1 has been updated with these new spectra. Fortunately, all conclusions remain the same. Spectra collected at lower temperatures have been relegated to the detailed methods and electronic supplement data file.

2. *The authors observed a solid precipitate (termed as 'uranium phase' in line 92 of the manuscript) after the sample was heated at 146 °C for 16 h, but didn't provide any further characterization. Please add some photographs of the solid precipitate in the supplementary material. Also, I suggest the authors to conduct some further scanning electron microscopic (SEM) observations and characterize its composition with energy diffraction spectrometer (EDS) or if possible, conduct some micro-XRD analysis. After conducting further characterization of the solid precipitate, the involvement of carbonate, silica, or uranyl can be better constrained.*

While the precipitate could be visually observed, good quality photographs of it could not be taken hence why we have elected not to show them. The relatively low concentration of uranium in solution also precluded any non-synchrotron based in situ XRD as the small total mass of precipitating solid and X-ray absorption by surrounding water would have made it very difficult to measure a usable diffraction pattern. SEM-EDS measurements of the solid were also infeasible as the solid invariably dissolved back into solution upon cooling. Thermodynamic modelling provides an alternative means of determining the solid's composition, our model suggests $UO_2(OH)_2(s)$ though if $Na_2U_2O_7$ is added to the model it is a possible alternative. A brief discussion on identifying the solid phase has been added around Lines 109-113.

3. *Did you observe any precipitates in your XAS experiments?*

Precipitates were observed in the experiments (at the chamber walls) but could not be characterised due to extremely small volumes.

4. *The authors have conducted some thermodynamic calculations to derive the pH and some other properties of the fluid. But some calculation results are not listed in the experimental data. Please provide more details of the thermodynamic calculations (including composition-pressure-temperature conditions, species distributions, pH, activities coefficients and etc.) in the files.*

Full calculation details have been added to the Supplementary Material Data

5. *In the solubility experiments, the author viewed the total amount of CO_3^{2-} (sourcing from $NaHCO_3$) as dissolved amount of carbonate. Can carbonate precipitates form in the system during heating? In addition, I noticed that the $\log_a[CO_3^{2-}]$ is much lower than $\log_m[CO_3^{2-}]$. Does that mean the activity coefficient is very low at such conditions (below 10⁻³)? Please list more detailed data of the thermodynamic calculations (also in question 4).*

We believe that new Raman data presented in the revised manuscript contradict this assumption. As it can be seen from the peaks evolution heating results in decomposition of uranyl-carbonate complexes and in building up

uncomplexed carbonate in the solution. In addition, prior thermodynamic calculations did not predict any carbonate phase precipitation.

With regards to the activity of carbonate the reviewer is correct but this does not alter the final conclusions, relevant data have been added to the Supplementary Material as mentioned in the response to Item 4.

6. *There are some format errors in the manuscript and supplementary material. Some sources didn't show in the supplementary material (line 137, line 239, line 249, line 254, line 257 in the supplementary material). The multiplicative sign '20×' should be used in front of the objective lens instead of '20X' (line 10, in the supplementary material). A space is needed between '16' and 'h' to represent 16 hours (lines 24, 29, 36, and 40 in the supplementary material). '-' and '–' should be clearly and consistently used in the main text, references, and figures! Please correct!*

Corrections implemented as suggested.

7. *The figures need further optimization. For example, the font of temperature marked in figures 3 and 4 are too larger than the tick labels and the title of the x/y axis.*

Corrections implemented as suggested

8. *Both fluid temperature and pressure should be specified in the main text.*

Corrections implemented as suggested

9. *Lin 75, (m denotes moles of solute ...)*

Correction implemented as suggested

10. *For HDAC experiments, how to determine the composition of the loaded fluid sample. In my opinion, the ultimate composition is different from that in the bulk solution. During sample loading, evaporation will result in an increase in the fluid concentration. In addition, a fluid pressure of 350 MPa is much higher than the geological counterparts.*

This is generally correct though we feel a slight increase in the concentration of all the solution's components does not change the conclusions drawn from the data we present. We agree that the HDAC pressures were a bit high but pressure typically has little effect on speciation behaviour and the diminished prevalence of carbonate complexes and ultimate precipitation observed in the HDAC experiment was consistent what was observed in the Raman experiments. So in this particular case we feel little further elaboration is required.

Reviewer 3

This paper is a systematic experimental study, employing an impressive variety of complementary techniques, of the effect of carbonate complexes on the solubility of uranyl hydroxide, (UO₂(OH)₂(s)), in sodium carbonate and bicarbonate solutions as the temperature is raised from ambient to hydrothermal conditions (250°C, psat). Raman spectroscopy and XAS using hydrothermal diamond anvil cells, composition-dependent solubility studies, and chemical equilibrium modelling are used to demonstrate that the uranyl ion, UO₂(2+) does not form stable carbonate complexes at these temperatures and that its solubility is governed by hydrolysis reactions under these experimental conditions. This finding has significant implications for some types of geological repositories for nuclear waste management and hydrothermal ore body formation. The experiments are very well designed and appear to have been carefully carried out. The discussion and conclusions are appropriate, and the topic is noteworthy.

Thanks

Suggested Changes/Corrections

The paper is well written, and I have only few specific suggestions:

- *The paper is somewhat verbose for a Chemical Communication. The text could be more succinct.*

Unfortunately, we could not identify any ways to shorten the text without removing important points or information. Indeed, feedback from other reviewers necessitated further additions to the text.

- *The captions to the figures need to be more carefully worded. Figures 3 and 4 should specify the mineral whose solubility is plotted.*

Alterations have been made as suggested

- *Lines 125 to 130. Was the equilibrium solid phase shown to be uranyl hydroxide, (UO₂(OH)₂(s)), or could it have transformed into uraninite, UO₂(s), or a uranyl carbonate mineral?*

The phase was determined to be UO₂(OH)₂ based on both thermodynamic calculations and post experimental XRD measurements of the solids used – an additional line has been added to the text stating this.

- *Line 136 and elsewhere: The term “uranium solubility” is imprecise. The saturating solid phase should be identified.*

We have now clarified the term as “UO₂(OH)₂ solubility” (Lines 166, Figure 3 and 4 captions).

Reviewers' comments:

Reviewer #1 (Remarks to the Author):

The authors have satisfactorily answered previous comments but several issues remain, as follows:

1. Manuscript

How can the authors prove that in their solubility experiments they did not measure the solubility of a secondary carbonate phase that might have formed during the experiments? If such a phase would form no change in U concentration with increasing carbonate would be expected. This would explain the "constant" U concentration in Figure 3. While in Figure 4 the solubility changes as a function of pH.

Also, in the new spectra the authors provided, the carbonate species do not "disappear" anymore but they seem to decrease due to a decrease in the solid phase solubility, thus a decrease of total dissolved U. Not necessarily a decrease in the stability of the carbonate complex.

L101 "Spectra were collected at 25, 50, 98 and 146 °C at 1 bar or water vapour pressure (whichever was higher)," – spectra collected at 25 and 90 are not shown in the manuscript nor in the supplementary material. In Figure 1 is written 150 C instead of 146 C.

L114 "at 146 °C we observed a loss of uranyl tricarbonate from solution (Figure 1)" – this is better termed as a decrease since there is still uranyl tricarbonate seen. Also, the observed changes are due to the precipitation of $\text{UO}_2(\text{OH})_2(\text{s})$ which leads to a decrease in dissolved uranium and an increase in the free carbonate species (less uranium to form complexes with carbonate).

L143 "effectively disappear from the solution when temperature exceeds 100-120 °C," – in figure 1 the species are still in solution at 150 C. Also in the XAS spectra one can still see the shoulder of the species – they don't disappear but they decrease in concentration.

Figure 3 – make the same Y axis for both figures

Could it be that in the solubility experiments you might had a uranium carbonate phase precipitating and that you are measuring its solubility and not that of $\text{UO}_2(\text{OH})_2$?

Figure 3 – shouldn't you plot "a log activity of uranium complex vs log activity of carbonate ion plot" - as written on line 171 instead of vs $\log m(\text{U})$ as is plotted in the actual Figure?

2. Probably due to a mistake, the supplementary material seems not to have been updated / corrected. The file in the system still contains a lot of missing reference errors, unrecognized characters, table without number on page 5.

Other issues in the supplementary material file are:

"after exposure extended exposure to temperature"

"These were our findings for spectra measured at up to 100 °C, data may be found in the electronic supplement data file." - copy paste from main manuscript without adapting the text.

"chemical system definition and activity model, both of which we have described above." – but no such information found above.

Reviewer #2 (Remarks to the Author):

The authors did a very good job in the revision. The manuscript looks much better now, both in science and readership. This is a very nice experimental work on the transport of uranium in hydrothermal fluids. I think the experimental data and thermodynamic description will provide

useful information for the study of U-mineralization process. I am now very glad to recommend its acceptance after minor corrections.

Lines 4-6: in the author list, full given names or initials? In addition, add spaces before Caporuscio and Matteo, respectively.

Lines 96, 143 and 174: variables, such as T and P should be in italic in the text.

Line 164: P-T conditions.

Lines 163 and 334: m; m is a length unit.

Line 276: H+.

Lines 279-281: Before reaching this conclusion, one might need to evaluate the relative stability of chloride, sulfate and hydroxyl uranyl complexes.

Lines 315-337: in methods of Raman spectroscopy, is the solution preparation procedure the same as that presented for the XAS experiments?

Reviewer #3 (Remarks to the Author):

Assessment of Revised Manuscript

As noted in my original review, this paper is a systematic experimental study, employing an impressive variety of complementary techniques, of the effect of carbonate complexes on the solubility of uranyl hydroxide, $(\text{UO}_2(\text{OH})_2(\text{s}))$, in sodium carbonate and bicarbonate solutions as the temperature is raised from ambient to hydrothermal conditions (250°C, psat). Raman spectroscopy and XAS using hydrothermal diamond anvil cells, composition-dependent solubility studies, and chemical equilibrium modelling are used to demonstrate that the uranyl ion, $\text{UO}_2(2+)$ does not form stable carbonate complexes at these temperatures and that its solubility is governed by hydrolysis reactions under these experimental conditions. This finding has significant implications for some types of geological repositories for nuclear waste management and hydrothermal ore body formation. The experiments are very well designed and appear to have been carefully carried out. The discussion and conclusions are appropriate, and the topic is noteworthy.

The comments by Reviewers 1 and 2 on the original manuscript are valid. The authors' corrections and edits in the revised manuscript address my comments and those of Reviewers 1 and 2 adequately.

Recommendation:

The paper is extremely well written and makes an important contribution to the literature. It should be published, as-is or with very minor editorial revisions.

Reviewer #1 (Remarks to the Author):

The authors have satisfactorily answered previous comments but several issues remain, as follows:

1. Manuscript

How can the authors prove that in their solubility experiments they did not measure the solubility of a secondary carbonate phase that might have formed during the experiments? If such a phase would form no change in U concentration with increasing carbonate would be expected. This would explain the "constant" U concentration in Figure 3. While in Figure 4 the solubility changes as a function of pH.

Reviewer 1 is correct in asserting that the formation of a uranyl carbonate phase at elevated temperatures could explain the lack of an observed carbonate-uranium solubility dependence. We believe this is unlikely as any such solid should be preserved during the quenching stage of the solubility technique – even if such a phase was very soluble at room temperature the separation of solid and solution after quenching would preclude its redissolution. We have added a brief discussion of this possibility in the main text ~Lines 195-200 and a new figure (Figure 4) in the Supplementary Methods and Results that hopefully better illustrates the whole solubility technique.

Also, in the new spectra the authors provided, the carbonate species do not "disappear" anymore but they seem to decrease due to a decrease in the solid phase solubility, thus a decrease of total dissolved U. Not necessarily a decrease in the stability of the carbonate complex.

Retrograde solubility is indeed a potential means to explain a decrease of uranium dissolved in solution. However, if retrograde solubility was the sole cause of this behaviour then it is likely that we would have seen a relationship between solubility and carbonate concentration in the solubility experiments, thus we maintain that the decrease of uranyl-carbonate in solution observed in the Raman and XAS experiments is largely a result of the destabilisation of uranyl carbonate complexes though retrograde solubility of $\text{UO}_2(\text{OH})_2$ may also contribute to a minor degree. We have added a brief discussion of this possibility around Lines 201-205.

L101 "Spectra were collected at 25, 50, 98 and 146 °C at 1 bar or water vapour pressure (whichever was higher)," – spectra collected at 25 and 90 are not shown in the manuscript nor in the supplementary material. In Figure 1 is written 150 C instead of 146 C.

Requested data have been added to the Supplementary Methods and Results. Thanks for catching the error in Figure 1, adjustment implemented

L114 "at 146 °C we observed a loss of uranyl tricarbonate from solution (Figure 1)" – this is better termed as a decrease since there is still uranyl tricarbonate seen. Also, the observed changes are due to the precipitation of $\text{UO}_2(\text{OH})_2(\text{s})$ which leads to a decrease in dissolved uranium and an increase in the free carbonate species (less uranium to form complexes with carbonate).

Wording has been adjusted as per your recommendation (Now Line 109)

L143 "effectively disappear from the solution when temperature exceeds 100-120 °C," – in figure 1 the species are still in solution at 150 C. Also in the XAS spectra one can still see the shoulder of the species – they don't disappear but they decrease in concentration.

True, wording has been adjusted to take this into account (Now Line 139)

Figure 3 – make the same Y axis for both figures

A good idea, adjustment implemented

Could it be that in the solubility experiments you might had a uranium carbonate phase precipitating and that you are measuring its solubility and not that of $\text{UO}_2(\text{OH})_2$?

Our response to your first comment addresses this point

Figure 3 – shouldn't you plot "a log activity of uranium complex vs log activity of carbonate ion plot" - as written on line 171 instead of vs $\log m(\text{U})$ as is plotted in the actual Figure?

Technically yes, and activities are all taken into account during numerical fitting and derivation of DeltaG in OptimA – OptimA models the whole system and accounts for all species in solution as well as their activities and optimises their deltaGs – this is how the properties for UO_2OH^+ were derived. The logm graphs were used as a preliminary analysis step in this process and present the data just as effectively as a loga graph – the only real difference between the two would be a vertical offset.

2. Probably due to a mistake, the supplementary material seems not to have been updated / corrected. The file in the system still contains a lot of missing reference errors, unrecognized characters, table without number on page 5.

Apologies for this, it is unclear what occurred here, it's possible we didn't upload the corrected version or some strange errors manifested during file transfer between authors. Hopefully this time everything should work properly.

Other issues in the supplementary material file are:

"after exposure extended exposure to temperature"

"These were our findings for spectra measured at up to 100 °C, data may be found in the electronic supplement data file." - copy paste from main manuscript without adapting the text.

Corrections implemented, although the line about 'electronic supplement data file' is in fact referring to the excel file that contains all our data, we couldn't think of a better way to refer to it.

"chemical system definition and activity model, both of which we have described above." – but no such information found above.

Bad wording on our part – the sentence should have referred back to the main text where the chemical system definition is described. This sentence has been corrected to make more sense. (~Lines 85-86 in supplementary material)

Reviewer #2 (Remarks to the Author):

The authors did a very good job in the revision. The manuscript looks much better now, both in science and readership. This is a very nice experimental work on the transport of uranium in hydrothermal fluids. I think the experimental data and thermodynamic description will provide useful information for the study of U-mineralization process. I am now very glad to recommend its acceptance after minor corrections.

Lines 4-6: in the author list, full given names or initials? In addition, add spaces before Caporuscio and Matteo, respectively.

Well spotted with the spaces, something of a surprise that we forgot to include full names in the author list. We have added full names where they were missing. Don't know if they will actually be used though.

Lines 96, 143 and 174: variables, such as T and P should be in italic in the text.

We have checked over all such variables in the text, and believe they are all in italics.

Line 164: P-T conditions.

Corrected as suggested

Lines 163 and 334: m; m is a length unit.

Corrected as suggested, though on our version the m at ~Line 163 was already in italics

Line 276: H+.

Seems to already be written as such, on our version it is shown as H^+

Lines 279-281: Before reaching this conclusion, one might need to evaluate the relative stability of chloride, sulfate and hydroxyl uranyl complexes.

True, hence the suggestion that a more detailed characterisation of uranyl hydroxyl speciation be carried out in order to permit an accurate evaluation of such relative stabilities. We have added a slight extension to the discussion emphasising this point. (~Lines 287-290)

Lines 315-337: in methods of Raman spectroscopy, is the solution preparation procedure the same as that presented for the XAS experiments?

Solutions were prepared differently for Raman and XAS experiments, we have added the solution preparation method to the Raman section. (Lines 342-343)

Reviewer #3 (Remarks to the Author):

Assessment of Revised Manuscript

As noted in my original review, this paper is a systematic experimental study, employing an impressive variety of complementary techniques, of the effect of carbonate complexes on the solubility of uranyl hydroxide, $(\text{UO}_2(\text{OH})_2(\text{s}))$, in sodium carbonate and bicarbonate solutions as the temperature is raised from ambient to hydrothermal conditions (250°C, psat). Raman spectroscopy and XAS using hydrothermal diamond anvil cells, composition-dependent solubility studies, and chemical equilibrium modelling are used to demonstrate that the uranyl ion, $\text{UO}_2(2+)$ does not form stable carbonate complexes at these temperatures and that its solubility is governed by hydrolysis reactions under these experimental conditions. This finding has significant implications for some types of geological repositories for nuclear waste management and hydrothermal ore body formation. The experiments are very well designed and appear to have been carefully carried out. The discussion and conclusions are appropriate, and the topic is noteworthy.

The comments by Reviewers 1 and 2 on the original manuscript are valid. The authors' corrections and edits in the revised manuscript address my comments and those of Reviewers 1 and 2 adequately.

Recommendation:

The paper is extremely well written and makes an important contribution to the literature. It should be published, as-is or with very minor editorial revisions.

Thanks!

REVIEWERS' COMMENTS:

Reviewer #1 (Remarks to the Author):

The authors satisfactorily addressed the previous comments. The paper is of interest to the scientific community and can be published, it provides relevant experimental data and discussions on uranium speciation at elevated temperatures in carbonate rich fluids.